# Crossing the threshold of ultrafast laser writing in bulk silicon

Margaux Chanal[1], Vladimir Yu. Fedorov[2,3], Maxime Chambonneau [1], Raphaël Clady[1], Stelios Tzortzakis [2,4,5] & David Grojo[1]

An important challenge in the field of three-dimensional ultrafast laser processing is to achieve permanent modifications in the bulk of silicon and narrow-gap materials. Recent attempts by increasing the energy of infrared ultrashort pulses have simply failed. Here, we establish that it is because focusing with a maximum numerical aperture of about 1.5 with conventional schemes does not allow overcoming strong nonlinear and plasma effects in the pre-focal region. We circumvent this limitation by exploiting solid-immersion focusing, in analogy to techniques applied in advanced microscopy and lithography. By creating the conditions for an interaction with an extreme numerical aperture near 3 in a perfect spherical sample, repeatable femtosecond optical breakdown and controllable refractive index modifications are achieved inside silicon. This opens the door to the direct writing of three-dimensional monolithic devices for silicon photonics. It also provides perspectives for new strong-field physics and warm-dense-matter plasma experiments.

[1] Aix-Marseille University, CNRS, LP3 UMR 7341, 13009 Marseille, France. [2] Science Program, Texas A&M University at Qatar, P.O. Box 23874, Doha, Qatar. [3] P. N. Lebedev Physical Institute of the Russian Academy of Sciences, 53 Leninskiy Prospekt, 119991 Moscow, Russia. [4] Institute of Electronic Structure and Laser (IESL), Foundation for Research and Technology—Hellas (FORTH), P.O. Box 1527, GR-71110 Heraklion, Greece. [5] Materials Science and Technology Department, University of Crete, 71003 Heraklion, Greece. Correspondence and requests for materials should be addressed to S.T. (email: stzortz@iesl.forth.gr) or to D.G. (email: grojo@lp3.univ-mrs.fr)

ntense femtosecond lasers are today used in many technolo-
gical and scientific domains ranging from precision material
processing[1, 2] and nanosurgery[3, 4] to the study of extreme non-
equilibrium conditions in solid state and plasma physics[5, 6].
However, three-dimensional precision applications based on
tightly focused beams in the bulk of matter remain so far limited
to large band gap transparent materials, like dielectrics[1, 5–7]
and some polymers[8, 9]. It is today commonly accepted that the
strong nonlinearities inherent to narrow-gap materials prevent
highly localized energy deposition of ultrafast infrared light in
the bulk[10, 11]. This, in turn, prevents achieving ultrafast
optical breakdown and/or local structural manipulations inside
important materials, such as silicon and other semiconductors.
Confirming this statement, recent attempts to achieve ultrafast
breakdown in the bulk of silicon have simply failed for the highest
pulse intensity and for all, even the largest, numerical aperture
(NA) tested so far[10–13]. Despite, or because of, this difficulty,
there are a few groups who recently turned to multipulse
accumulation regimes[13, 14] or nanosecond pulse durations[15–17].
While bulk modifications in silicon have been widely reported in
the latter case, the thermal nature of such interactions does not
allow attaining a level of control comparable to that demonstrated
for femtosecond breakdown inside dielectrics, making such
approaches unrealistic for high-precision applications. Thus,
achieving controllable ultrafast modifications in semiconductors
today still appears to be the most promising approach.

In this article, we carefully examine the limitations inherent to
infrared femtosecond interactions in semiconductors to fully
identify their origins. We demonstrate the absence of a strict
physical limit to ultrafast energy localization, which allows us to
detail an experimental solution to achieving permanent refractive
index control in the bulk of silicon. Inspired by solid-immersion
microscopy, this solution is based on hyper-focused ultrafast
beams that are intrinsically free from aberrations leading to
highly confined interactions deep into silicon.

## Results

**Flat target interactions**. To experimentally investigate highly
localized interactions inside high-resistivity crystalline silicon
samples, which are opaque in the visible range of the spectrum,

one needs to specifically develop accurate infrared microscopy
techniques. A simplified schematic of our methodology is shown
in Fig. 1a. In the experiments, 60-fs laser pulses at 1300-nm
wavelength are tightly focused below the surface of the samples to
create nanosecond-lived[18] microplasmas initiated by two-photon
absorption[19] (2PA) at the focus. By imaging these microplasmas,
we first reveal a saturation of the maximum electron density that
can be achieved at $\cong 5 \times 10^{19}$ cm$^{-3}$, which is about one order of
magnitude below the critical plasma density[11]—the most widely
applied criterion for ultrafast optical breakdown and subsequent
modification[20]. While it is natural to associate this observation
to an optical limitation on the laser intensity that can be
delivered, we provide here direct experimental evidence by
three-dimensional (3D) propagation imaging.

A few examples of the measurements are given in Fig. 1b,
where cross-sections of fluence distributions are displayed
for pulses focused with a NA of 0.45. The low energy case
(1 pJ, $P \cong 16$ W) lies far below all thresholds for nonlinear
effects and the linear point spread function (PSF) of our
arrangement is measured. At laser energies above the 2PA
threshold ($\cong 10^{10}$ W cm$^{-2}$ corresponding to $\cong 1$ nJ[19]) we observe a
nonlinear transformation of the PSF progressively taking a carrot-
like shape with the maximum delivered fluence moving to the
pre-focal region. All displayed distributions are normalized to
their maxima provided in Fig. 2a with additional measurements
(see *hollow symbols*). These measurements naturally reveal an
increase of the delivered fluence with NA. However, a
striking feature is also the quenching in all cases at a level
significantly below the material modification threshold measured
at $0.35 \pm 0.03$ J cm$^{-2}$ (horizontal line) with the same laser focused
on the sample surface (see Supplementary Note 1). As
bulk modification thresholds are usually higher than surface
thresholds[21], we decided to take this observable as a minimal
fluence target to achieve bulk structural modifications.

One should note that the recorded fluence images in Fig. 1 are
limited in resolution by the 0.7 NA of the imaging objective
and consequently we limit our measurements to a maximum
NA of 0.65 (for pump focusing). To investigate higher NA values,
we inevitably needed to turn to nonlinear propagation modeling.
In principle, the nonparaxial nature of our problem demands the

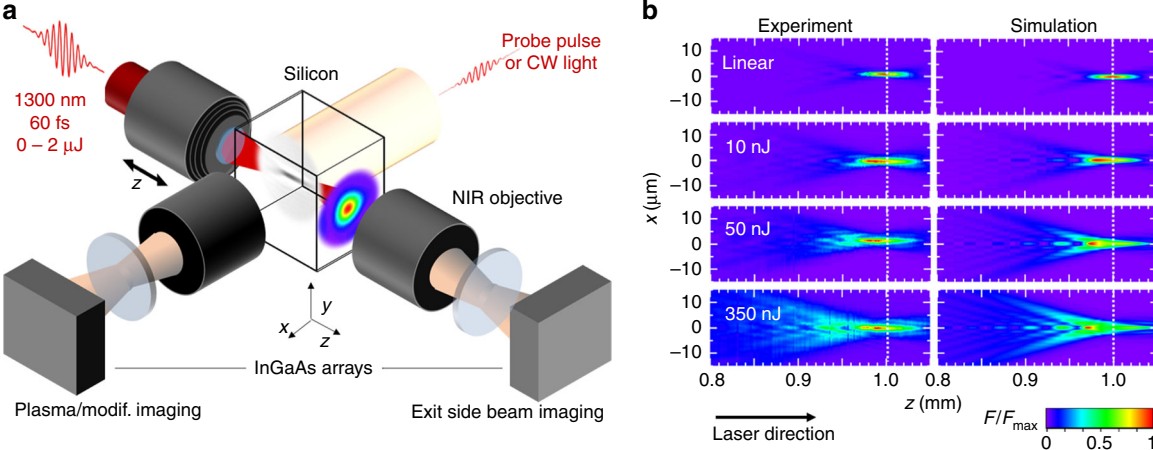

**Fig. 1** Accessing ultrafast laser energy and plasma density inside silicon. **a** Simplified schematic of the measurement methods: Two customized infrared
microscopes, each composed of an objective, a tube lens and an InGaAs camera, are positioned laterally and along the optical axis. Ultrashort probe pulses
illuminate the interaction region for lateral imaging of transient microplasmas and permanent modifications. Beam profiling at the exit surface of the sample
is performed to retrieve, by a z-scanning procedure, the 3D distributions of the delivered laser fluence inside silicon. **b** Cross-sections of fluence
distributions for ultrashort pulses focused in silicon with a numerical aperture of 0.45: The beam focus is positioned at a depth of 1 mm inside silicon as
shown by the *dotted white lines*. Measurements and simulations are compared for increasing pulse energies up to 350 nJ as labeled for each row (the label
"linear" stands for a sub-nanojoule energy ensuring the absence of nonlinear effects). All distributions are normalized to their maximum fluence $F_{max}$

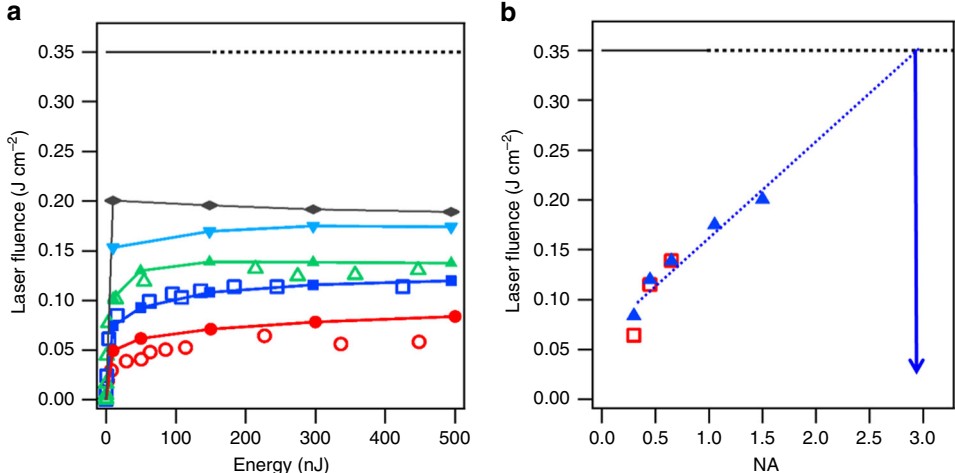

**Fig. 2** Maximum delivered laser fluences inside silicon with ultrashort pulses. Measurements (*hollow symbols*) and simulations (*solid symbols*) for evaluating the peak fluence delivered with 60-fs laser pulses focused at a depth of 1 mm inside silicon. **a** Dependence on the incoming laser pulse energy evaluated for increasing numerical apertures (NA): NA = 0.3 (*red*), NA = 0.45 (*dark blue*), NA = 0.65 (*green*), NA = 1 (*light blue*), and NA = 1.5 (*gray*). **b** Fluence saturation levels plotted as a function of NA. Taken as a target, the *dark horizontal line* is the measured threshold for modification with the same laser beam at the surface of the sample. The *blue arrow* indicates by extrapolation that a NA near 3 is required to cross this fluence threshold in the bulk

use of Maxwell vectorial solvers that suffer from the need of unrealistically large computational resources. To overcome this issue, we used our recently demonstrated transformation optics approach to treat the problem with simple nonlinear scalar wave equations[22], accounting for Kerr-induced nonlinearities, dispersion, 2PA, collision-assisted ionization and plasma effects. Figures 1b and 2 show the simulated fluences for femtosecond laser pulses at 1300-nm wavelength propagating in silicon. These compare favorably with the measurements showing that the essential contributions to the intensity clamping are correctly accounted in the model. Performing parametric numerical studies (see Supplementary Note 4), we identified that a first contribution to the clamping is the strong and progressive depletion of the pulse energy, well before reaching the focus. This depletion is due to the high 2PA coefficient inherent to narrow-gap materials. For instance, it is striking to note that <20% of a 500-nJ pulse reaches the focal region in this experiment. However, other important contributions are plasma absorption and defocusing[23]. As the critical plasma density scales inversely to the wavelength, we found that these effects induce already large losses at very low energies (typ. 20 nJ) in comparison to more conventional experiments in dielectrics at visible wavelengths.

For circumventing these effects, tight focusing is a natural option to lower the intensity and interaction length in the pre-focal region. However, it is important to highlight here the drawback of processing the interior of a high index material as silicon ($n_{Si} = 3.5$) because refraction at the interface leads to a relatively paraxial energy flux for all cases. For instance, for NA near 1, the half-maximum angle in silicon, directly given by $\theta = \text{asin}(NA/n_{Si})$ is less than 17 degrees. This low apparent NA explains the modest benefits observed when repeating the simulations for NA = 1 and NA = 1.5 (Fig. 2a), the typical maximum value for oil-immersion objectives.

More precisely, we observe in Fig. 2b that the maximum fluence that can be delivered increases with NA and a linear extrapolation of the obtained data (*dashed line*) shows that extreme NA values near 3 are required to exceed the originally targeted fluence. Such extreme NA definitively require a technological disruption. In that perspective, we identified that the drawback of a high index could be transformed into an advantage by applying the solid-immersion strategy, originally proposed for resolution enhancement in microscopy[24]. The

principle is rigorously the same as liquid-immersion microscopy except that the liquid is replaced by a solid of higher refractive index to further increase the apparent NA. Because objectives are designed to transform the incoming plane wave into a perfect spherical wave (truncated at half angle $\theta = \text{asin}(NA)$), we can use silicon directly as the solid-immersion medium, provided that the flat samples are replaced by silicon spheres and the radiation is focused at their exact centres. This way totally suppresses refraction at the sample interface and the apparent NA is directly the NA in air multiplied by $n_{Si} = 3.5$ (see Supplementary Note 2 for details). In other words, the focusing inside spheres using a NA = 0.3 objective is already tighter (effective NA > 1) than previously tested configurations.

**Spherical target experiments**. To demonstrate experimentally that the concept of 3D ultrafast laser writing in bulk silicon with hyper-NA values holds, we have focused our pulses at the centre of 1-mm-radius silicon spheres for direct comparison with previous measurements at a depth of 1 mm inside silicon wafers. Then, using a set of near-infrared objectives with specified NA values up to 0.85, experiments in silicon spheres correspond to effective NA values up to 2.97. At such extreme NA, the spot size shrinks down to about $\lambda/(2NA) \cong 220$ nm, which is far below the spatial resolution of any of our optical diagnostics. Thus, we decided to simply concentrate on the detection of potential permanent material modifications using the same lateral microscopy set-up as the one used for microplasma observations (see Fig. 1a). As predicted, no permanent modification could be detected for NA values up to 2.25 (even under repeated illumination with high energy pulses of 10 µJ). According to Fig. 2a, we expected that the maximum fluence is already delivered with pulses of only 20-nJ energy for all cases. Thus, we decided to investigate our highest NA possibility (effective NA = 2.97) at this low energy level. Figure 3a shows infrared microscopy images of the focal region after irradiation with increasing number of pulses. While it is hardly resolved due to imaging limitations, a modification is successfully detected even with the first applied shot (see Fig. 3b showing the image in logarithmic scale). Under repeated illumination, the modification first assumes the focal volume (10 pulses) and expands, with exposure, preferentially in the prefocal region. These results

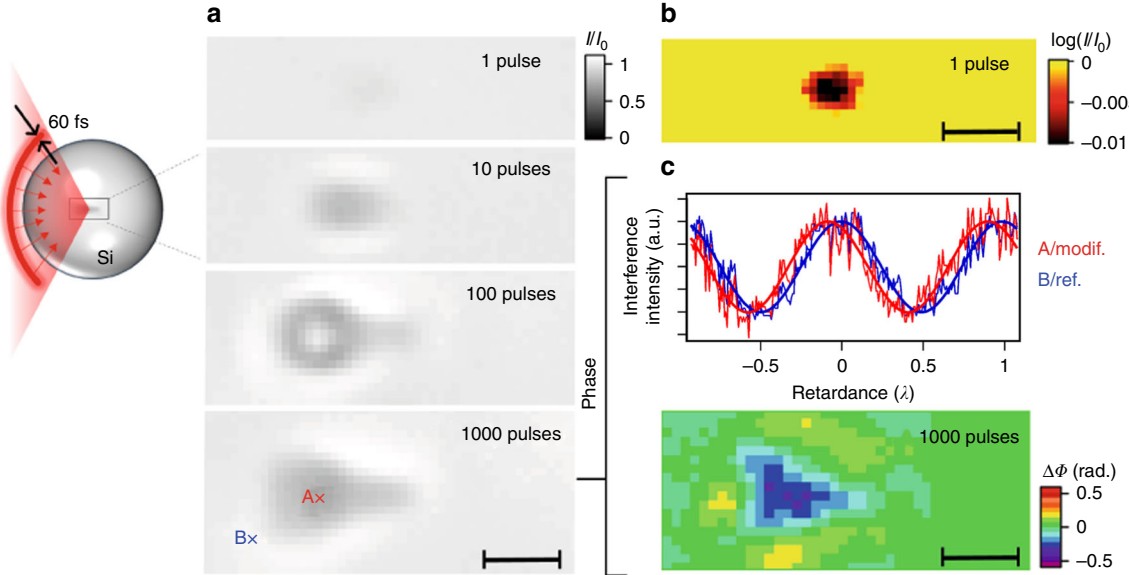

**Fig. 3** Refractive index modification achieved in the bulk of silicon with ultrashort pulses. Micro-modifications are created at the centre of silicon spheres using focused 60-fs laser pulses with the hyper-NA value of 2.97 (illustrated by the sketch) and then analyzed by infrared microscopy. **a** Bright-field infrared images of modifications for different number of applied laser pulses. **b** Transmission image in logarithmic scale revealing a modification after the first applied laser shot. **c** Differential-longitudinal interferometry measurement (for two pixels A and B shown on the image for 1000 pulses) and corresponding phase image for the modification created with 1000 applied pulses. This indicates a local change of the silicon refractive index $\Delta n < -0.07$ after repeated illumination. (*scale bars*: 2 μm)

demonstrate for the first time a local modification in the bulk of silicon achieved with sub-100-fs laser pulses. We have also repeated the experiment in several silicon spheres with different material grades. Systematically repeatable observations confirmed that the demonstration is not based on a nondescript level of impurities and that the pulse energy of 20 nJ exceeds the breakdown threshold significantly.

For technological considerations, it is natural to be immediately interested in the refractive index changes associated with this ultrafast modification. These changes can be measured by applying a longitudinal-differential interferometry methodology with our microscopy experiment (see Supplementary Note 3 for details). A reference wave is superimposed on the array sensor for interference with the imaging wave. Then, by simply varying with nanometer precision the optical path difference between the waves, we can extract the phase and refractive index variations (see A and B traces in Fig. 3c corresponding to positions given in Fig. 3a for 1000 pulses). By repeating a simplified procedure on each pixel of the images (4-step phase shifting), we show in Fig. 3c the reconstruction of a phase image for the modification by 1000 applied laser pulses. According to the axial symmetry of the structure around the optical axis and the modification diameter $d = 1.2$ μm, the maximum phase change $\Delta\Phi$ that was observed corresponds to a negative index variation $\Delta n = \Delta\Phi \cdot \lambda / (2\pi \cdot d) < -0.07$. The index change at the first laser shot was below the sensitivity limit of our in situ infrared phase diagnostic. This suggests an ultrafast material response similar to dielectrics with an athermal structural change caused bond-breaking due to ionization[25] that progressively transforms in material nanodisruptions under repeated illumination, and associated with the measurement of large negative index variations[26, 27]. This result opens a direct way to 3D laser refractive index engineering in silicon by the same way as it is today commonly achieved in wide band-gap materials. The rarefied material region, which is observed is obviously associated with the compression of the material against the surrounding crystal forming a densified region, which ultimately can form

super-dense phase materials via fast quenching. Interestingly, this is already visible on the phase image shown in Fig. 3c where one can note three regions with positive index variation around the triangular-shape modification.

## Discussion

We extrapolate that an even higher NA would have been required for material modification with shorter pulses and/or longer wavelengths. However, there is a limit in NA, given by the material refractive index that will be hardly exceeded. Our demonstration opens the door to solid-immersion lens technologies for 3D laser writing in silicon. These technologies already hold promises in microscopy[24, 28, 29] and lithography[30, 31] for which solutions for 3D control of aberration-free focal spots have been demonstrated using special solid-immersion lens assemblies[29] and adaptive optics concepts[32]. However, they are so far very rarely adopted due to their practical complexity and the availability of other resolution enhancement methods. Interestingly, these advances find in silicon laser writing an application where the complexity is fully justified as we report a solution for 3D ultrafast laser refractive-index engineering inside silicon. Silicon is the backbone material for micro- and nano-technologies but it is striking to note that our demonstration of local refractive-index modification in the bulk of silicon arrives only 20 years later after those in wide band gap dielectrics[33, 34]. Looking at the future, it seems feasible to push the concept further by borrowing another microscopy technique: the 4-Pi arrangement[35] that would consist in interacting with two (or even more) crossing hyper-NA focused pulses at the centre of spheres. This may open new possibilities to achieve warm-dense-matter conditions in semiconductors and the so-called micro-explosion experiments that are today limited to materials with wide gaps[6].

## Methods

**Ultrafast laser interactions**. The experiments are carried out with a Titanium:Sapphire laser source (LP3-ASUR facility) combined with an optical parametric

amplifier (HE-TOPAS, Light Conversion) that delivers pulses which can exceed 1-mJ energy at 1300-nm wavelength. The pulses have a duration of 60 fs as measured using a single-shot autocorrelator (TiPA, Light Conversion). For all experiments, the linearly polarized pulses are tightly focused at a depth of 1 mm inside silicon by using microscope objectives from Olympus LCPLN-infrared series, which are designed for semiconductor system infrared inspections. These objectives of NAs ranging from 0.1 to 0.85 are equipped with an adjustable collar for correction of the spherical aberration due to silicon thickness. For all investigated conditions, the correction is adjusted (as described in the Supplementary Note 2) so that we only deal with linear PSFs that are free from spherical aberration. The depth of 1 mm is chosen to ensure that the laser fluence on the surface is low ($<$GW cm$^{-2}$) for all tested situations and there is no material damage or surface effects influencing the bulk measurements.

**Three-dimensional laser fluence imaging**. The beam focus analysis is conducted using the on-axis imaging system as shown in Fig. 1a. A microscope objective with NA higher than that for focusing (NA = 0.7 Mitutoyo near-infrared series) is used to image the focal spot at the exit surface (back) of the sample onto an InGaAs camera (Raptor, OWL SWIR 640). The camera directly measures the beam fluence profiles according to the linear response of the detector at 1300 nm and a calibration procedure consisting in a focus analysis at low intensity in air (linear propagation—see Supplementary Note 1). The focusing objective is mounted on a motorized stage, which is indexed at 0.5 μm corresponding to $\cong$1.75 μm between images due to refractive index mismatch at Air/Si interface (z-scan procedure). Because there is no material after the imaged plane and the intensities of the measured beam are low enough to avoid any nonlinear propagation effects in air, we are dealing only with real space images. Accordingly, we directly reconstruct from the collected stack of images the full 3D fluence distributions in silicon. For all acquisitions, the laser repetition rate is 100 Hz and the exposure time is 40 ms so that each profile relies on an average over four laser pulses. The spatial resolution of this diagnostic is obviously subject to the diffraction limit of our NA = 0.7 observing objective. For these reasons, pump focusing with the NA = 0.85 objective is not investigated by this method. By comparing the measured distributions with simulations, we estimate the resolution and dynamic of the measurements are about 1 μm and 19 dB, respectively.

**Infrared imaging of microplasmas and modifications**. We use a pump and probe lateral microscopy setup to measure by free-carrier absorption the characteristics of the microplasmas induced inside silicon with our pump pulses. The experimental arrangement based on a second InGaAs array detector (XENICS, XEVA 1.7-640) is similar to that for the study of the fast kinetics of free-carriers injected in bulk silicon by TPA[11]. For microplasma observations, the optical delay between the pump and probe pulses is set at only 10 ps so that we observe the free-carrier distributions before any significant decay[18]. For the detection and characterization of permanent modifications in the silicon spheres, the pump is blocked after the illumination. For phase microscopy of the modifications, we rely on a longitudinal-differential interferometry technique[36] translated in the infrared domain of the spectrum. Phase images are achieved by using two identical 20×-magnification microscope objectives in both arms of the interferometer (Olympus LCPLN20XIR, NA = 0.45). All experimental details and data are provided in the Supplementary Note 3. For amplitude bright-field imaging we replace the ultrafast probe by a non-coherent infrared illumination (quartz-tungsten halogen lamp) for improved bright-field imaging performance.

**Samples**. For the microplasma and fluence imaging experiments, we use microelectronics grade high-resistivity silicon crystals (Float zone silicon, orientation (100) < 0.5°, resistivity >900 Ω cm corresponding to an initial free-electron density $<5 \times 10^{12}$ cm$^{-3}$). For microplasma studies, the targets are $10 \times 10 \times 10$ mm$^3$ cubes prepared with optical polishing on all sides (Crystal GmbH) so that there is no sample edge diffracting the probe in the field of view. Microplasmas are formed and observed at a depth of 1 mm below the surface facing the pump beam. For the 3D fluence measurements, the target is changed into a wafer of 1-mm thickness (Siltronix) which is mounted perpendicular to the optical axis. By z-scanning the pump beam focus across the back surface of the wafer, the fluence distribution is reconstructed from the collected images as if the focus were delivered 1-mm inside a silicon slab, similarly to the microplasma imaging experiment. For experiments with hyper-focused pulses, we use 2-mm diameter spheres made of High Resistivity Float Zone Silicon (resistivity >10 kΩ cm, Tydex).

**Nonlinear propagation modeling**. To simulate the interactions, we solve a Unidirectional Pulse Propagation Equation coupled to a free-carrier density rate equation. Despite the nonparaxial nature of these situations, we use a scalar treatment based on the transformation optics approach, detailed and demonstrated in the reference[22]. The solved equations, the parameters applied in the simulations and the results of the parametric analysis to identify the factors that prevent reaching the material modification threshold in conventional experiments are given in the Supplementary Note 4.

**Data availability**. The data that support the findings of this study are available from the corresponding authors upon reasonable request.

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

## Acknowledgements

We thank O. Utéza and A. Rode (Australian National Univ.) for helpful discussions, and J.L. Bellemain for his technical contribution on the sphere experiment. This research has been performed in the frame of the International Associated Laboratory "MINOS". The project has been supported by the National Priorities Research Program grant No. NPRP9-383-1-083 from the Qatar National Research Fund (member of The Qatar Foundation), the European Union's Horizon 2020 research and innovation programme grant agreement No 724480 from the European Research Council (ERC), and the A*MIDEX project (ANR-11-IDEX-0001-346 02) funded by the "Investissements d'Avenir," French Government program, managed by the French National Research Agency (ANR).

## Author contributions

S.T., D.G. designed the research. Mar.C., Max.C., R.C. performed the experiments. V.Y.F. developed the numerical model and performed the theoretical study. All authors interpreted the results and contributed to the manuscript prepared by D.G.

## Additional information

**Competing interests:** The authors declare no competing financial interests.

