## [Peer Review File · Nature Communications]

Reviewers' comments:

Reviewer #1 (Remarks to the Author):

The manuscript presents an analysis of the possible ways of volumetric modification of silicon by IR femtosecond laser pulses. This includes sophisticated experiments with varying laser energy and numerical aperture of laser beam focusing. The experiments are supported by numerical modeling of laser beam propagation with laser-generated electron plasma. It has been shown that, due to high refractive index of silicon, volumetric modification cannot be reached in single-pulse irradiation regimes even at numerical apertures above 1. Finally, an original solution on silicon volumetric modification is proposed based on using millimeter-sized spheres. As a whole, the manuscript is very well written, considers the problem under solution in details and from different viewpoints, and is expected to be of high interest for laser-matter interaction and material processing communities.

The manuscript can be recommended for publishing in Nature Communications with minor revision:

1. In abstract, it is recommended to change "A remaining challenge" to "One of remaining challenges" or even "An important challenge" otherwise it looks that almost all problems of 3D laser processing of materials have been solved.
2. It is also recommended to refer readers to supplementary information in several places in the manuscript, where appropriate, e.g., line 78 (surface damage threshold).
3. The authors state that "repeatable femtosecond optical breakdown and controllable refractive index modifications are achieved for the first time inside silicon". It must be underlined that it is achieved in the regime of trains of single pulses. With double femtosecond laser pulses, modification of silicon has been achieved by Shimotsuma et al. (JLMN, 11(1), 35 (2016)). This paper has to be cited in the manuscript with a corresponding comment.
4. Have the authors performed an analysis of silicon modification at zones of increased refractive index? Can they comment about modified silicon structure? Can it be layered as in the above-mentioned paper by Shimotsuma?

Reviewer #2 (Remarks to the Author):

A. Summary of the key results: This work continues the exploration of laser writing inside silicon. Here, 60 fs laser pulses with a wavelength of 1300 nm are used for deep subsurface irradiation of silicon (depth of 1 mm) with air NA objectives. At these conditions, light losses due to two photon absorption and plasma defocusing, analyzed by means of optical imaging and simulations, impel the strong confinement of light in the focal volume required to surpass the threshold for silicon modification. To overcome this issue, the solid immersion effect, provided by a 2mm silicon sphere, is used, and a change in refractive index of -0.07 is shown.

B. Originality and significance: As far as I am aware, this work is novel. However, there are several works that show the structuring of bulk silicon using nanosecond laser pulses. They are relevant to put this work into perspective, and they should also be cited: "Crystal structure of laser-induced subsurface modifications in Si", P.C. Verbug et al., Appl. Phys. A, 120 (2015) 683–691; "Writing waveguides inside monolithic crystalline silicon with nanosecond laser pulses", M. Chambonneau et al., Opt. Lett., 41 (2016) 4875;" In-chip microstructures and photonic devices fabricated by nonlinear laser lithography deep inside silicon", O. Tokel et al., arXiv:1409.2827 (2014).

C. Data & methodology: validity of approach, quality of data, quality of presentation

3D writing: The key limitation of the use of a silicon sphere for light confinement is how to implement this approach for 3D writing. Thus, the authors should clarify the sentence from line 180: "This results opens a direct way to 3D laser refractive index engineering in silicon...". One potential solution could be the integration of the SIL in an AFM, as shown Ref. 24 ("Near-field photolithography with a solid immersion lens", L.P. Ghislain et al., Appl. Phys. Lett. 74 (1999)). However, damage of the SIL is difficult to prevent in this case. Alternatively, it is possible to write nanopatterns in an area close to the SIL center, as shown in "Sub-wavelength Laser Nanopatterning using Droplet Lenses", M. Duocastella et al., Sci. Reports 5 (2015) 16199. Still, how to control the z position of the focal volume for 3D writing remains for the authors to be explained.

Thickness: Related to the last point, two photon absorption and plasma defocusing are strongly dependent on the z position of the focus inside silicon. All experiments reported in the current work consider 1 mm depth, but it would also be interesting to perform simulations at a lower depth, a more realistic position considering the thickness of silicon wafers to be usually about 500 μm .

Alignment: The aplanatic condition of a SIL can be lost if the center of sphere is not properly positioned at the focal plane of the objective lens (as mentioned in section 3 in the Supplementary Material). This can also limit the implementation of the technique. Discussion on this issue is required.

D. Suggested improvements

Fig. 1b: The fluence distributions in Fig. 1b are said to be normalized. Thus, I suggest to indicate so in the colorbar of this figure.

Fig. 2b: Only 3 data measurements are indicated in this figure, but at least 5 values were experimentally measured (as shown in Fig. 2a). Also, does the value at a NA close to 3 correspond to a calculated value? If not, why do the authors assume a linear extrapolation?

3D laser fluence measurements: When axially moving the objective to measure the 3D fluence in steps of 500 nm, I assume that the authors accounted for the shift in the focus position caused by the refractive index mismatch. It would be instructive to add this information.

Supplementary Table 1: The values for the half angle seem a little bit off (e.g. for 0.3 in air, the angle should be $\text{asin}(0.3)=17.5$, and not 17.9). What was the procedure to calculate them?

Supplementary Fig. 5b: A scale bar is missing.

E. References: The references mentioned above should be included.

F. Clarity and context: In general, the manuscript is clear and the writing is of a high standard.

It has long been conjectured that the strong optical nonlinearity and high refractive index of silicon prevent the concentration of laser energy in the silicon bulk to the level sufficient enough to induce optical breakdown and permanent modification of crystal structure.

The paper “Crossing the threshold of ultrafast laser writing in bulk silicon” shows an elegant way to break this paradigm. It demonstrates the way to suppress the defocusing of laser radiation due to the very high refractive index of silicon (~ 3.5) by matching precisely the curvature of the target surface and the spherical converging wavefront in the high-NA focusing optics. That completely suppressed spherical and coma aberrations. As a result, the deposited energy density into the material (in J/cm^3) has been increased by at least an order of magnitude. A permanent modification of Si by ultrafast laser has been clearly demonstrated and characterised using longitudinal-differential interferometry methodology. The manuscript represents an important advance in laser interaction with matter which, to my opinion, should be published. I expect the presented results will attract significant attention from a broad range of specialists in laser micromachining, 3D-microstructuring and silicon photonics. I thus strongly recommend this work for publication in Nature Communications.

However, I would like to make a few suggestions and ask authors a couple of questions.

- First, at the end of the first introduction paragraph (lines 44 – 46, p.2) I would recommend to add a sentence explaining how did you break the limitation in silicon. Something like this: “We eliminate defocusing of the laser radiation in silicon by matching the converging spherical wavefront of the laser pulse with a spherical target surface of the same curvature.” In the current version this is not clearly stated anywhere in the text, and becomes obvious only by close examination of Fig.3 in p.6.
- Second, you have clearly observed modification of Si by a single laser pulse (see p.6, Fig.3b.). Could you estimate the deposited energy density in J/cm^3 to induce such changes?
- Third, Please add scale in Figs.3b and c at the bottom, or indicate in the figure capture that the scale is the same as in images Fig.3a.
- Fig. 3a. – Schematics: I would suggest to add a description of the experimental setup and add that this picture illustrate the way to eliminate spherical and coma aberrations and illuminate the defocusing of the beam by matching the curvature of the converging wavefront and the curvature of the spherical target at the surface.
- Fig.3a – again. The images clearly show the growth of a ‘tail’ of the damaged spot in the direction of laser pulse propagation. As you use the linearly polarised pulses, could this be an indication of the z-component of the laser field and the resulted formation of a ‘needle’ beam? Could you comment on this?
- Following the title of the paper, could you indicate the threshold which you are crossing from the results of your experiments (in J/cm^3)?
- I would recommend adding some in-line subtitles, such as, for example: “Flat target experiments.” (in italic) at the beginning of line 47, p.2; and “Spherical target experiments” at the beginning of line 130, p.5.

Summing up, I would like to congratulate the authors with a very nice experimental results and well presented manuscript. I really enjoyed reading the manuscript and the supplementary materials to the manuscript.

Reviewer #1 (Remarks to the Author):

The manuscript presents an analysis of the possible ways of volumetric modification of silicon by IR femtosecond laser pulses. This includes sophisticated experiments with varying laser energy and numerical aperture of laser beam focusing. The experiments are supported by numerical modeling of laser beam propagation with laser-generated electron plasma. It has been shown that, due to high refractive index of silicon, volumetric modification cannot be reached in single-pulse irradiation regimes even at numerical apertures above 1. Finally, an original solution on silicon volumetric modification is proposed based on using millimeter-sized spheres. As a whole, the manuscript is very well written, considers the problem under solution in details and from different viewpoints, and is expected to be of high interest for laser-matter interaction and material processing communities.

The manuscript can be recommended for publishing in Nature Communications with minor revision:

1. In abstract, it is recommended to change “A remaining challenge” to “One of remaining challenges” or even “An important challenge” otherwise it looks that almost all problems of 3D laser processing of materials have been solved.

Our response:

We agree and we have replaced “*A remaining challenge in the important field ...*” by “*An important challenge in the field ...*”

2. It is also recommended to refer readers to supplementary information in several places in the manuscript, where appropriate, e.g., line 78 (surface damage threshold).

Our response:

We acknowledge that not all Supplementary items were cited in the main text. In the revised version of our manuscript, we have divided the Supplementary information in 4 Supplementary Notes and made sure that each Note is cited in the main manuscript appropriately. For instance, we refer to Supplementary Note 1 when we compare the bulk measurements with the surface damage threshold.

3. The authors state that “repeatable femtosecond optical breakdown and controllable refractive index modifications are achieved for the first time inside silicon”. It must be underlined that it is achieved in the regime of trains of single pulses. With double femtosecond laser pulses, modification of silicon has been achieved by Shimotsuma et al. (JLMN, 11(1), 35 (2016)). This paper has to be cited in the manuscript with a corresponding comment.

Our response:

Following this recommendation, also made by referee #2, we have reformulated the introduction to cite the previous studies reporting modifications in the long-pulse and pulse-train regimes (Shimotsuma et al, Tokel et al., Verburg et al., Chambonneau et al.).

4. Have the authors performed an analysis of silicon modification at zones of increased refractive index? Can they comment about modified silicon structure? Can it be layered as in the above-mentioned paper by Shimotsuma?

Our response:

Interestingly, we report a negative index change while longer pulse regimes or pulse-train regimes have reported positive refractive index modification so far. This directly evidences a specificity of the ultrafast regime. This result suggests an ultrafast silicon response, similar to that of dielectrics with an athermal structural change caused by

bond-breaking that progressively transforms in material nanodisruptions under repeated illumination, and associated with the measurement of large negative index variations. While this view was mentioned in the original manuscript (l 180), we did not highlight that the rarefied region is obviously associated with the compression of the material against the surrounding crystal forming a densified region, which ultimately can form super-dense phase materials via fast quenching. Interestingly, this is already visible in the phase image of Fig 3c where one can note 3 regions with positive index variation around the triangular-shape modification. We added a comment on this observation in the text of the revised version (last two sentences of the “Results” section). However, more details on the silicon structure in these regions would require X-Ray or electron diffraction analyses that are complex for such small regions (<1µm) and even more so in the bulk of silicon. While we will concentrate our future efforts on these aspects, we cannot conclude on the silicon structure at the moment and we prefer to more reasonably confine our conclusion on the potential of our scheme for the study of new dense phases of silicon. This appears in the final sentence: “*This may open new possibilities to achieve warm-dense-matter conditions in semiconductors and the so-called micro-explosion experiments that are today limited to dielectrics*”

Reviewer #2 (Remarks to the Author):

A. Summary of the key results: This work continues the exploration of laser writing inside silicon. Here, 60 fs laser pulses with a wavelength of 1300 nm are used for deep subsurface irradiation of silicon (depth of 1 mm) with air NA objectives. At these conditions, light losses due to two photon absorption and plasma defocusing, analyzed by means of optical imaging and simulations, impel the strong confinement of light in the focal volume required to surpass the threshold for silicon modification. To overcome this issue, the solid immersion effect, provided by a 2mm silicon sphere, is used, and a change in refractive index of -0.07 is shown.

B. Originality and significance: As far as I am aware, this work is novel. However, there are several works that show the structuring of bulk silicon using nanosecond laser pulses. They are relevant to put this work into perspective, and they should also be cited: “Crystal structure of laser-induced subsurface modifications in Si”, P.C. Verbug et al., Appl. Phys. A, 120 (2015) 683–691; “Writing waveguides inside monolithic crystalline silicon with nanosecond laser pulses”, M. Chambonneau et al., Opt. Lett., 41 (2016) 4875;” In-chip microstructures and photonic devices fabricated by nonlinear laser lithography deep inside silicon”, O. Tokel et al., arXiv:1409.2827 (2014).

Our response:

Following this recommendation, also made by referee #1, we have reformulated the introduction to cite the previous studies reporting modifications in the long-pulse and pulse-train regimes (Shimotsuma et al, Tokel et al., Verbug et al., Chambonneau et al.).

C. Data & methodology: validity of approach, quality of data, quality of presentation
3D writing: The key limitation of the use of a silicon sphere for light confinement is how to implement this approach for 3D writing. Thus, the authors should clarify the sentence from line 180: “This results opens a direct way to 3D laser refractive index engineering in silicon...”. One potential solution could be the integration of the SIL in an AFM, as shown Ref. 24 (“Near-field photolithography with a solid immersion lens”, L.P. Ghislain et al., Appl.

Phys. Lett. 74 (1999)). However, damage of the SIL is difficult to prevent in this case. Alternatively, it is possible to write nanopatterns in an area close to the SIL center, as shown in “Sub-wavelength Laser Nanopatterning using Droplet Lenses”, M. Duocastella et al., Sci. Reports 5 (2015) 16199. Still, how to control the z position of the focal volume for 3D writing remains for the authors to be explained.

Our response:

On rereading our paper with this comment in mind, we realised that we confined the practical perspectives for 3D laser writing to only one sentence at the end of the paper: *“Our demonstration opens the door to solid immersion lens (SIL) technologies, which already hold promises in microscopy and lithography but are very rarely adopted due to their practical complexity and the availability of other resolution enhancement methods. Interestingly, it finds here an application where the complexity is fully justified as we report the sole solution, identified to date, for 3D ultrafast laser modification inside silicon.”* We acknowledge this may appear short in communicating what is our vision of the suitable schemes for 3D control.

Expanding on this, we have demonstrated the validity of SIL scheme for bulk silicon modification by a proof-of-concept experiment in a perfect silicon sphere. Although we would like to be able to provide a more complete demonstration with SIL scheme directly compatible with 3D scanning and the realization of complex photonic structures, such a realization as you point out, will require further technological advances that take time.

You mention two potential technological solutions and we agree that both are limited for laser writing applications either due to the potential damage of the SIL or the difficulty for 3D scanning of the beam focus. We can however propose other solutions, which are also directly inspired by advanced microscopy developments. In particular, we can mention the recent work by Agarwal, K et al. “Crossing the resolution limit in near-infrared imaging of silicon chips: Targeting 10-nm node technology”. Phys. Rev. X 5, 1–9 (2015) in which an ingenious assembly is described for the purpose of holding and accurately aligning a hemispherical silicon sample (the SIL), pressing it onto a flat sample (to be 3D processed) to avoid an air gap between the SIL and the sample and finding the correct focal plane for aberration free focusing. While the authors concentrate on resolution enhancement for microscopy, their design could be directly transposed for our laser writing demonstration as an effective NA of 3.3 is obtained for a sphere diameter of 3 mm preventing any damage issue (NA is 3 for a diameter of 2 mm in our demonstration). The ultrahigh resolution 2D images shown in this paper directly demonstrate the XY scanning suitability of such a scheme but we agree that Z-scanning remains a challenge that will require an extra technological complication. To achieve axial-scanning, one may consider adaptive optics solutions as the Z-position of the focus can be directly adjusted by dynamical control of the wavefront of the beam incoming to the SIL. To illustrate this possibility, we can refer to another advanced confocal microscopy study by Mudry, E et al. “Isotropic diffraction-limited focusing using a single objective lens.” Phys. Rev. Lett. 105, 1–4 (2010) in which a phase-only spatial light modulator was used for shaping the incident beam in order to focus simultaneously at two-points and to Z-scan the focus for 3D imaging. One could also refer to a similar implementation for a different application: Optical trapping with 3D control. (see for instance: Rodrigo, et al. “Real-time three-dimensional optical micromanipulation of multiple particles and living cells.” Opt. Lett. 29, 2270–2272 (2004).)

In our revised manuscript, we have introduced these ideas and references in the “Discussion” section of the paper to place more clearly our results in this technological context. However, we prefer to keep this discussion at the level of general concepts, as

different technology directions can be considered to address this issue. Our view is very simple: we have found a long looked-after solution to this exciting problem and the purpose of this paper is to make the scientific community aware of it so that these technological advances for practical applications will be coming much faster.

Thickness: Related to the last point, two photon absorption and plasma defocusing are strongly dependent on the z position of the focus inside silicon. All experiments reported in the current work consider 1 mm depth, but it would also be interesting to perform simulations at a lower depth, a more realistic position considering the thickness of silicon wafers to be usually about 500 μm .

Our response:

We have actually examined this problem. Experiments give similar results but for simplicity we chose 1-mm depth to avoid any surface effect. As can be seen in the simulation, it is true that two photon absorption leads to a progressive depletion (see Supplementary Fig. 9). For this reason, we predict that depths much smaller than 500 μm would limit the losses due to this absorption. However, the same simulations also reveal that plasma effects that give the main contribution to the intensity clamping are significant only in the focal region (about 50 μm before the focus). This directly evidences the very modest dependence on the depth (provided it exceeds about 100 μm). To address this point, we comment on the localization of the plasma effects in the Supplementary Note 4 concentrating on simulation details.

Alignment: The aplanatic condition of a SIL can be lost if the center of sphere is not properly positioned at the focal plane of the objective lens (as mentioned in section 3 in the Supplementary Material). This can also limit the implementation of the technique. Discussion on this issue is required.

Our response:

We believe that a dynamical control of the incoming wavefront using a SLM is a natural potential solution to this issue. For this reason, we mention the aberrations issue when describing in the “Discussion” section the required technological advances for practical applications (see above).

D. Suggested improvements

Fig. 1b: The fluence distributions in Fig. 1b are said to be normalized. Thus, I suggest to indicate so in the colorbar of this figure.

Our response:

For more clarity, we have modified the label of the colorbar into “F/F_max” and introduced “F_max” the maximum fluence found for each distribution in the figure legend. This information is also repeated in the text: “All displayed distributions are normalized to their maxima provided in figure 2a with additional measurements (see hollow symbols).”

Fig. 2b: Only 3 data measurements are indicated in this figure, but at least 5 values were experimentally measured (as shown in Fig. 2a). Also, does the value at a NA close to 3 correspond to a calculated value? If not, why do the authors assume a linear extrapolation? 3D laser fluence measurements: When axially moving the objective to measure the 3D fluence in steps of 500 nm, I assume that the authors accounted for the shift in the focus position caused by the refractive index mismatch. It would be instructive to add this information.

Our response:

We checked the consistency between measurements and simulations. There are 3 measurements (3 different NAs up to 0.65) in Fig 1a shown by hollow symbols and reported in Fig 1b also shown by hollow symbols. Here it is important to note that the fluence distributions that are measured are potentially low-pass filtered, by our 0.7 numerical aperture imaging objective. For this reason, we do not expect to resolve all details and we have limited the measurements to a maximum NA value of 0.65 (for pump focusing). To investigate higher NAs we turned to modeling that allowed us to add values for NA=1 and NA=1.5. While we have demonstrated the validity of our simulation approach for such large NA values, we haven't repeated the simulations for higher NAs because the extreme nonparaxial nature of the problem would have raised new questions on the validity of the calculations, which are out of the scope of this paper. While there is no physical argument to directly justify a linear extrapolation, it is striking to note that it predicts impressively well the silicon modification threshold.

For clarity on these aspects, we replaced the sentence on page 3: *“Due to infrared imaging limitations, we inevitably needed to turn to nonlinear propagation modeling to investigate extreme-NA values.”* by *“One should note that the recorded fluence images in Fig 1 are limited in resolution by the 0.7 numerical aperture of the imaging objective and consequently we limit our measurements to a maximum NA value of 0.65 (for pump focusing). To investigate higher NA values we inevitably needed to turn to nonlinear propagation modeling.”*

We also confirm that index mismatch is accounted in the shift of beam focus by 500 μm increments of the objective. This information is added in the “methods” section.

Supplementary Table 1: The values for the half angle seem a little bit off (e.g. for 0.3 in air, the angle should be $\text{asin}(0.3)=17.5$, and not 17.9). What was the procedure to calculate them?
Supplementary Fig. 5b: A scale bar is missing.

Our response:

We apologize for a systematic small error in the values reported in this table. We confirm the angles are simply estimated according to $\text{asin}(\text{NA})$ or $n_{\text{Si}}\text{asin}(\text{NA})$ depending on the situations. We have corrected all erroneous values in this table.

E. References: The references mentioned above should be included.

Our response:

We have included these references (see above).

F. Clarity and context: In general, the manuscript is clear and the writing is of a high standard.

Reviewer #3 (Remarks to the Author):

It has long been conjectured that the strong optical nonlinearity and high refractive index of silicon prevent the concentration of laser energy in the silicon bulk to the level sufficient enough to induce optical breakdown and permanent modification of crystal structure.

The paper “Crossing the threshold of ultrafast laser writing in bulk silicon” shows an elegant way to break this paradigm. It demonstrates the way to suppress the defocusing of laser radiation due to the very high refractive index of silicon (~ 3.5) by matching precisely the curvature of the target surface and the spherical converging wavefront in the high-NA

focusing optics. That completely suppressed spherical and coma aberrations. As a result, the deposited energy density into the material (in J/cm³) has been increased by at least an order of magnitude. A permanent modification of Si by ultrafast laser has been clearly demonstrated and characterised using longitudinal-differential interferometry methodology. The manuscript represents an important advance in laser interaction with matter which, to my opinion, should be published. I expect the presented results will attract significant attention from a broad range of specialists in laser micromachining, 3D-microstructuring and silicon photonics. I thus strongly recommend this work for publication in Nature Communications.

However, I would like to make a few suggestions and ask authors a couple of questions.

- First, at the end of the first introduction paragraph (lines 44 – 46, p.2) I would recommend to add a sentence explaining how did you break the limitation in silicon. Something like this: “We eliminate defocusing of the laser radiation in silicon by matching the converging spherical wavefront of the laser pulse with a spherical target surface of the same curvature.” In the current version this is not clearly stated anywhere in the text, and becomes obvious only by close examination of Fig.3 in p.6.

Our response:

We added at the end of the introduction the sentence:” *Inspired by solid-immersion microscopy, this solution is based on hyper-focused ultrafast beams that are intrinsically free from aberrations leading to an unprecedented level of confinement deep into silicon.*”

- Second, you have clearly observed modification of Si by a single laser pulse (see p.6, Fig.3b.). Could you estimate the deposited energy density in J/cm³ to induce such changes?

Our response:

We acknowledge that it is a question of major interest but it will be hardly answered for such a complex scheme and such a confined interaction. It is true that all our discussions conclude on a minimum laser fluence (J/cm²) to be delivered to achieve modification. Our conclusion is that we obtain a modification in bulk when targeting a fluence exceeding the fluence threshold for surface modification (about 0.35 J/cm²). The main difficulty for reporting a threshold in energy density in J/cm³ is that it would require performing a precise 3D characterization of the laser energy distribution at the focus, which is a very challenging task under these experimental conditions (for the extreme-NA value of 3). While we have concentrated on achieving the first ultrafast laser modification in silicon and their potential technological benefits, the underlying material science is only partially understood at the present time. More focused studies will follow together with efforts on the tentative enhancement of energy deposition up to a microexplosion regime. In such cases the deposited energy density is the essential parameter that can be retrieved from analysis of the size of the created void and the shock-affected zones (see for instance Vailionis, A. *et al.* [“Evidence of superdense aluminium synthesized by ultrafast microexplosion”. *Nat. Commun.* 2, 445 (2011).]

- Third, Please add scale in Figs.3b and c at the bottom, or indicate in the figure capture that the scale is the same as in images Fig.3a.

Our response:

This information is added in the figure caption. “The scale shown in the bottom left image applies for all images.”

- Fig. 3a. – Schematics: I would suggest to add a description of the experimental setup and add that this picture illustrate the way to eliminate spherical and coma aberrations and illuminate the defocusing of the beam by matching the curvature of the converging wavefront and the curvature of the spherical target at the surface.

Our response:

We acknowledge that the details of the experimental configuration to break the limitation in silicon appear relatively late in the paper. This is because the solution is built on the careful examination of the conventional situations and which represents the core of our demonstration. We believe that the schematic view of the sphere interactions given in Fig. 3 is enough to illustrate the basis of our solution. However, to address this remark, we also state more clearly the solution that is described in the introduction (see response above). We refer also more clearly to the Supplementary Note 2 entitled “Focusing and imaging in a Si-sphere” in which all relevant features for our solution are given.

- Fig.3a – again. The images clearly show the growth of a ‘tail’ of the damaged spot in the direction of laser pulse propagation. As you use the linearly polarised pulses, could this be an indication of the z-component of the laser field and the resulted formation of a ‘needle’ beam? Could you comment on this?

Our response:

This is an interesting comment that we actually did not consider in our analysis. However, the shape that we observe is not unusual for bulk interactions under repeated illumination. As for dielectrics, we mention in the manuscript that the modification first assumes the focal volume and expands, with exposure, preferentially in the prefocal region. Here, it is worth noting that the ‘tail’ shape appears only for more than 10 applied shots. While we are limited by the resolution of optical observation, the shape of the modification remains relatively spherical for a low number of pulses. We agree that the extreme-NA interactions of our study make reasonable the hypothesis of a significant z-component of the laser field at the focus but we believe that any discussion on a potential so-called ‘needle’ beam from these observations would be very speculative at this stage.

- Following the title of the paper, could you indicate the threshold which you are crossing from the results of your experiments (in J/cm³)?

Our response is given above (second comment).

- I would recommend adding some in-line subtitles, such as, for example: “Flat target experiments.” (in italic) at the beginning of line 47, p.2; and “Spherical target experiments” at the beginning of line 130, p.5.

Our response:

Following this suggestion, the “Results” section is now divided by subheadings: “Flat target interactions” and “Spherical target experiments”

Summing up, I would like to congratulate the authors with a very nice experimental results and well presented manuscript. I really enjoyed reading the manuscript and the supplementary materials to the manuscript.

REVIEWERS' COMMENTS:

Reviewer #1 (Remarks to the Author):

The authors convincingly answered to criticism and corrected the manuscript accordingly. I recommend this manuscript for publishing in Nature communications.

Reviewer #2 (Remarks to the Author):

In this revised version the authors have satisfactorily addressed all the comments from the reviewers. Therefore, I recommend the manuscript for publication in Nature Communications. I only have some minor suggestions to further improve the text:

-In the supplementary information, line 124, the equation that defines the confocal parameter is incorrect (otherwise, the focal volume would be almost isotropic!). The last term of the equation is missing a factor of 2. Most importantly, in order to consider that the beam is propagating in silicon, the expression should also be multiplied by a factor of 3 (refractive index of silicon). Thus, the confocal parameter should be about 660 nm.

-In the methods section, line 252, the sentence that starts with "The potentially diffracted fields..." is confusing. It seems that you intentionally low-pass filtered the images, whereas I believe that you simply mean that the 0.7 NA objective acts as a low-pass filter. Please rephrase.

-Details of the objective used for characterizing the plasma (or the modifications of the refractive index) are missing.

-The use of SILs in laser ablative processes has been kept at a minimum. One of the few that uses the immersion effect for direct laser writing is that by Duocastella et al., "Sub-wavelength laser nanopatterning using droplet lenses", *Sci. Reports* 5, 16199 (2015). I suggest adding this reference in the discussion section.

Reviewer #3 (Remarks to the Author):

I would like to thank the authors for incorporating all of the requested changes to their paper, answering all the questions and thereby clearly improving it. From my point of view, this work provides undoubtedly an important advance in the ultrafast laser interaction with the condensed matter, which thus should be published. I thus strongly recommend this work for publication in Nature Communications.

Reviewer #1 (Remarks to the Author):

The authors convincingly answered to criticism and corrected the manuscript accordingly. I recommend this manuscript for publishing in Nature communications.

Reviewer #2 (Remarks to the Author):

In this revised version the authors have satisfactorily addressed all the comments from the reviewers. Therefore, I recommend the manuscript for publication in Nature Communications. I only have some minor suggestions to further improve the text:

-In the supplementary information, line 124, the equation that defines the confocal parameter is incorrect (otherwise, the focal volume would be almost isotropic!). The last term of the equation is missing a factor of 2. Most importantly, in order to consider that the beam is propagating in silicon, the expression should also be multiplied by a factor of 3 (refractive index of silicon). Thus, the confocal parameter should be about 660 nm.

We apologize for this error on the confocal parameter that we have corrected in the Supplementary Note 2. We also checked throughout the documents that the error has not been repeated.

-In the methods section, line 252, the sentence that starts with “The potentially diffracted fields...” is confusing. It seems that you intentionally low-pass filtered the images, whereas I believe that you simply mean that the 0.7 NA objective acts as a low-pass filter. Please rephrase.

We acknowledge it may be confusing. We reformulate the sentence which becomes: “The spatial resolution of this diagnostic is obviously subject to the diffraction limit of our $NA=0.7$ observing objective.”

-Details of the objective used for characterizing the plasma (or the modifications of the refractive index) are missing.

These details were only available in the supplementary information. To clarify, we add in the Methods section the sentence: “Phase images are achieved by using two identical 20x-magnification microscope objectives in both arms of the interferometer (Olympus LCPLN20XIR, $NA=0.45$).”

-The use of SILs in laser ablative processes has been kept at a minimum. One of the few that uses the immersion effect for direct laser writing is that by Duocastella et al., “Sub-wavelength laser nanopatterning using droplet lenses”, Sci. Reports 5, 16199 (2015). I suggest adding this reference in the discussion section.

Following this suggestion, we have added this reference in the discussion section (see ref. 31)

Reviewer #3 (Remarks to the Author):

I would like to thank the authors for incorporating all of the requested changes to their paper,

answering all the questions and thereby clearly improving it. From my point of view, this work provides undoubtedly an important advance in the ultrafast laser interaction with the condensed matter, which thus should be published. I thus strongly recommend this work for publication in Nature Communications.